# Efficient Optimal PAC Learning

**Mikael Møller Høgsgaard**                                              HOGSGAARD@CS.AU.DK
*Aarhus University*

**Editors:** Gautam Kamath and Po-Ling Loh

## Abstract

The PAC model (Probably Approximately Correct) was introduced by Valiant (1984); Vapnik and Chervonenkis (1964, 1974) and has since been a cornerstone in machine learning theory. The PAC model is based on the idea that a learning algorithm, given labelled training examples, with high probability, should be able to learn how to label unseen examples with high accuracy. This means that the algorithm should generalize well from the training data to new data.

The most well-known PAC-learning setting is the binary realizable setting, where it is assumed that there exists an unknown true binary labelling function $c : \mathcal{X} \rightarrow \{-1, 1\}$, and an unknown distribution $\mathcal{D}$ over $\mathcal{X}$. The goal of the learner $\mathcal{A}$ is to determine, given $m$ labelled training examples $\mathbf{S} = ((\mathbf{x}_1, c(\mathbf{x}_1)), \ldots, (\mathbf{x}_m, c(\mathbf{x}_m)))$ by $c$ and $\mathbf{x}_1, \ldots, \mathbf{x}_m \sim \mathcal{D}$, a labelling rule $\mathcal{A}(\mathbf{S}) \in \{-1, 1\}^{\mathcal{X}}$ that minimizes the error $\mathcal{LD}(\mathcal{A}(\mathbf{S})) = \mathbb{P}_{x \sim \mathcal{D}}[\mathcal{A}(\mathbf{S})(x) \neq c(x)]$ on new points drawn from $\mathcal{D}$ and labelled by $c$.

In this setting, Vapnik and Chervonenkis (1974) and Blumer et al. (1989) showed that if there exists a hypothesis class $\mathcal{H} \subseteq \{-1, 1\}^{\mathcal{X}}$ of VC-dimension $d$ such that $c \in \mathcal{H}$, then, with probability at least $1 - \delta$ over random labelled training examples $\mathbf{S}$, all $h \in \mathcal{H}$ satisfying $\mathcal{LS}(h) = \frac{1}{m} \sum_{(x,y) \in \mathbf{S}} 1[h(x) \neq y] = 0$ also satisfy $\mathcal{LD}(h) \leq O(\frac{d \ln(m/d) + \ln(1/\delta)}{m})$. Thus, assuming $\mathcal{H}$ is known to the learner, the intuitive learning rule of empirical risk minimization (ERM), choosing any $h \in \mathcal{H}$ such that $\mathcal{LS}(h) = 0$, which is possible since the labelling function $c \in \mathcal{H}$, leads to the generalization error $\mathcal{LD}(\mathrm{ERM}(\mathbf{S})) = O(\frac{d \ln(m/d) + \ln(1/\delta)}{m})$.

From a lower bound perspective, Bousquet et al. (2020) showed that there exists a hypothesis class $\mathcal{H}$ of VC-dimension $d$, a target concept $c \in \mathcal{H}$, and a distribution $\mathcal{D}$ such that any proper learner $\mathcal{A}$, always outputting a hypothesis $\mathcal{A}(\mathbf{S}) \in \mathcal{H}$, with probability at least $1 - \delta$ has generalization error $\Omega(\frac{d \ln(m/d) + \ln(1/\delta)}{m})$. In contrast, for general learning rules $\mathcal{A}$ not restricted to outputting classifiers in $\mathcal{H}$, the lower bound on the generalization error is $\Omega(\frac{d + \ln(1/\delta)}{m})$ due to Ehrenfeucht et al. (1989).

Thus, a natural question is whether there exists a learner $\mathcal{A}$ achieving a generalization error of $\Theta(\frac{d + \ln(1/\delta)}{m})$. This question was positively answered in the seminal work of Hanneke (2016), which introduced the first optimal PAC learner. Hanneke's learner $\mathcal{A}$ uses a clever deterministic subsampling scheme that creates $\approx m^{0.79}$ sub-training sequences of size $\Theta(m)$ from $\mathbf{S}$. The output of $\mathcal{A}$ is a majority vote over $\approx m^{0.79}$ hypotheses, where each hypothesis results from running an ERM algorithm on one of the sub-training sequences, giving an efficient optimal PAC learner.

Moreover, seminal work by Larsen (2023) showed that bagging, introduced by Breiman (1996), also leads to an optimal PAC learner. Specifically, Larsen (2023) demonstrated that the majority vote of $\Theta(\ln(m/\delta))$ hypotheses, each obtained by running an ERM on a bootstrap sample of size $\Theta(m)$, is an optimal PAC learner and more efficient than the algorithm in Hanneke (2016).

We notice that both approaches require running an ERM algorithm on $\Theta(m)$ training examples. Thus, a natural question is whether it is possible to obtain an optimal PAC learner $\mathcal{A}$ that only calls a black-box ERM algorithm on $O(m)$ training examples.

---

*Extended abstract. Full version appears as (Høgsgaard, 2025).

In this work aim to answer this question, concretely we show that by calling the black-box ERM algorithm $O(\ln(\frac{m}{\delta(d+\ln(1/\delta))}) \ln(\frac{m}{\delta}))$ times, each time with $550d$ training examples, one can construct an optimal PAC learner as a majority vote over $O(\ln(\frac{m}{\delta(d+\ln(1/\delta))}))$ hypotheses. This new optimal PAC learner gives an alternative trade-off in computational cost compared to the efficient optimal PAC learner of Larsen (2023), which is more favorable if the ERM algorithm's computational cost scales poorly with the number of training examples.

**Keywords:** Binary Classification, Realizable PAC Learning, Boosting.

## Acknowledgments

We thanks Kasper Green Larsen for valuable conversations and his support. We also thank the anonymized reviewers for their comments helping improve the paper. Supported by Independent Research Fund Denmark (DFF) Sapere Aude Research Leader Grant No. 9064-00068B.

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
